# Brief communication: A clarification of wake recovery mechanisms

Maarten Paul van der Laan[1], Mads Baungaard[1], and Mark Kelly[1]

[1]Technical University of Denmark, DTU Wind Energy, Risø Campus, Frederiksborgvej 399, 4000 Roskilde, Denmark

**Correspondence:** Maarten Paul van der Laan (plaa@dtu.dk)

**Abstract.** Understanding wind turbine wake recovery is important for developing models of wind turbine interaction employed in the design of energy-efficient wind farm layouts. Wake recovery is often assumed or explained to be a shear-driven process; however, this is generally not accurate. In this work we show that wind turbine wakes recover mainly due to the divergence (lateral and vertical gradients) of Reynolds shear stresses, which transport momentum from the freestream towards the wake center. The wake recovery mechanisms are illustrated using a simple analytic model and results of large-eddy simulation.

## 1 Introduction

Wind turbine wakes can cause energy losses in wind farms and increase blade fatigue loads. Hence, understanding wind turbine wakes is important for designing energy-efficient wind farm layouts. Wake recovery is the process describing the flow's return to an undisturbed state via turbulent mixing. The wind energy science community (including the main author of the present work) often refers to the shear at the wake edges as the main driver behind the wake recovery, as the production of the turbulent kinetic energy depends on the square of the mean shear (van der Laan, 2014; Porté-Agel et al., 2020). Other authors have analyzed the mean kinetic energy budget of a wind farm using wind tunnel measurements (Cal et al., 2010; Newman et al., 2014) and large-eddy simulations (LES, e.g. Calaf et al., 2010; Andersen et al., 2017); they concluded that the vertical shear stress component of the Reynolds stress is the main driver behind energy transport of the freestream into the wake. Meyers and Meneveau (2013) computed transport tubes of the streamwise momentum and energy in wind farms using LES, and showed that the energy is transported sideways and top-down, where the dominant direction depends on the turbine lateral spacing. While the shear and the vertical Reynolds stresses are indeed important, they are not the precise reason why wake recovery occurs, since the Reynolds-averaged Navier-Stokes equation for streamwise momentum includes gradients of Reynolds stresses (stress divergence) that cause turbulent mixing. If a Reynolds stress is represented by a velocity gradient following the hypothesis of Boussinesq (1897), then it becomes clear that the *gradient of the shear* is responsible for wake recovery, and not the shear or Reynolds stresses themselves. This brief communication is meant to clarify the main mechanisms behind wake recovery, through use of a simple illustrative model of the far wake (Sect. 2), and by analyzing LES results (Sect. 3) to confirm the trends of the simple model.

## 2 A simple illustrative model of far wake recovery

The wind turbine wake can be split into near and far wake regions (Vermeer et al., 2003). The near wake is a result of the wind turbine blade forces and it is characterized by complex vortex structures that break down in to smaller turbulent eddies further downstream. The near wake velocity deficit is mainly a footprint of the wind turbine thrust force distribution and it diffuses downstream in a smoother velocity deficit profile. We define the far wake as the region where the mean velocity deficit has become self-similar. In other words, the far wake has forgotten how it was generated and only information of the

total wind turbine extracted momentum is known. We derive a simple model for the far wake with the aim of creating an illustrative example of the main wake recovery mechanism. The model is not meant to be used for the prediction of a wind turbine wake flow. We start with the Reynolds-averaged Navier-Stokes (RANS) momentum equation for incompressible and high Reynolds-number flow, for the streamwise direction:

$$\frac{DU}{Dt} = -\frac{1}{\rho}\frac{\partial P}{\partial x} - \frac{\partial \overline{u'u'}}{\partial x} - \frac{\partial \overline{u'v'}}{\partial y} - \frac{\partial \overline{u'w'}}{\partial z} + f, \tag{1}$$

where $U$ is the mean streamwise velocity, $\rho$ is the air density, $P$ is the mean pressure, $f$ is the wind turbine thrust force that we choose to represent by an actuator disk (AD) model (Réthoré et al., 2014), $t$ is the time and $x_j = (x, y, z)$ are the streamwise, lateral and vertical coordinates. The normal Reynolds stress $\overline{u'u'}$ and the shear Reynolds stresses $\overline{u'v'}$ and $\overline{u'w'}$ need to be modeled; we apply the well known hypothesis of (Boussinesq, 1897):

$$\overline{u'u'} = \frac{2}{3}k - 2\nu_T\frac{\partial U}{\partial x}, \qquad \overline{u'v'} = -\nu_T\left(\frac{\partial U}{\partial y} + \frac{\partial V}{\partial x}\right), \qquad \overline{u'w'} = -\nu_T\left(\frac{\partial U}{\partial z} + \frac{\partial W}{\partial x}\right), \tag{2}$$

with $k$ as the turbulent kinetic energy that can be absorbed in the pressure and $\nu_T$ as the eddy viscosity. In latter two expressions, we will neglect the $\partial/\partial x$ contributions to simplify the illustrative model.

Around the AD, a strong adverse pressure gradient is present that reduces the streamwise velocity upstream and downstream of the rotor. In absence of the Reynolds stresses, one can derive the well known 1D (axial) momentum solution for the streamwise velocity at the AD and at the far wake (Sørensen, 2016). The latter can be written as a velocity deficit, $\Delta U$, and

can be related to the thrust coefficient, $C_T$: $\Delta U/U_H = 1 - \sqrt{1 - C_T}$, with $U_H$ as the freestream velocity. In a turbulent flow, the divergence of the Reynolds stresses recover the streamwise velocity back to the freestream velocity. The 1D momentum solution for the velocity deficit can be seen as the maximum deficit that one could obtain in turbulent flow of an AD.

It can be shown that for zero pressure gradient and a constant eddy viscosity, the far wake velocity deficit is self-similar and can be modeled by a Gaussian function, as shown by Pope (2000). Bastankhah and Porté-Agel (2014); Xie and Archer (2015)

used wind tunnel measurements and large-eddy simulations of a wind turbine wake to show that the far wake velocity deficit can indeed be approximated by a Gaussian function:

$$\Delta U_{\text{wake}}(y, z) = \Delta U_{\text{max}} \exp\left[-\frac{\left(y^2 + [z - z_H]^2\right)}{2\sigma^2}\right] \tag{3}$$

where $\Delta U_{\text{max}}$ is the maximum deficit that is normally a function of the downstream distance but can be considered as a constant for fixed downstream position $x$, $z_H$ is the wind turbine hub height and $\sigma$ is the spatial scale (standard deviation) of

the Gaussian wake profile. We model the far wake velocity as a combination of a Gaussian velocity deficit and a logarithmic inflow similar to Bastankhah et al. (2021):

$$U(y,z) = U_{\text{in}}(z) - \Delta U_{\text{wake}}(y,z) = \frac{u_*}{\kappa} \ln\left(\frac{z}{z_0}\right) - \Delta U_{\text{wake}}(y,z), \tag{4}$$

where $U_{\text{in}}(z)$ represents a neutral atmospheric surface layer with $u_*$ as the friction velocity and $z_0$ as the roughness length. The shear stresses and their contribution to the momentum equation (known as stress divergence) become:

$$\overline{u'v'} = -\nu_T \frac{\partial U}{\partial y} = -\frac{\nu_T}{\sigma^2} y \Delta U_{\text{wake}}(y,z), \tag{5}$$

$$\overline{u'w'} = -\nu_T \frac{\partial U}{\partial z} = -u_*^2 - \frac{\nu_T}{\sigma^2}(z - z_H)\Delta U_{\text{wake}}(y,z) \tag{6}$$

$$-\frac{\partial \overline{u'v'}}{\partial y} = \nu_T \frac{\partial^2 U}{\partial y^2} = \frac{\nu_T}{\sigma^2}\left(1 - \frac{y^2}{\sigma^2}\right)\Delta U_{\text{wake}}(y,z), \tag{7}$$

$$-\frac{\partial \overline{u'w'}}{\partial z} = \nu_T \frac{\partial^2 U}{\partial z^2} + \frac{\partial U}{\partial z}\frac{\partial \nu_T}{\partial z} = \frac{\nu_T}{\sigma^2}\left(2 - \frac{z_H}{z} - \frac{[z - z_H]^2}{\sigma^2}\right)\Delta U_{\text{wake}}(y,z). \tag{8}$$

Here, we have assumed that the eddy viscosity is unaffected by the wake and equal to the logarithmic inflow: $\nu_T = u_* \kappa z$ by assuming a neutral atmospheric surface layer to be valid. This is a strong assumption and does not hold for non-neutral atmospheric conditions and for tall wind turbines that may operate beyond the surface layer. The assumption of a linear inflow eddy viscosity is the same as assuming a constant eddy viscosity in the far wake in order to derive a one-dimensional Gaussian profile as function of $y$, for each height $z$. The eddy viscosity of a real wind turbine far wake is expected to be non-uniform; RANS simulations of a single wind turbine wake typically show a Gaussian-like lateral eddy viscosity profile with its maximum in the wake center. Townsend (1949), Johansson et al. (2003) and Cafiero et al. (2020) have proposed modified Gaussian velocity deficit profiles to better match measured far wake results of a circular cylinder, axi-symmetric wake of a disk and an axi-symmetric wake of a plate, respectively, in order to account for the effect of non-uniform eddy viscosity on the self-similar velocity deficit. It should be noted that these measurements were performed for low Reynolds numbers that are three orders of magnitude lower than that of utility scale wind turbines (using $D = 100$ m), which makes their conclusions not directly applicable to our flow of interest. Therefore, RANS simulations of a single utility scale wind turbine are performed in Appendix A. The RANS simulations indicate that the wake-generated eddy viscosity has a minor impact on the far wake velocity deficit shape. The assumed Gaussian velocity profile of the simple model also results in self-similar shear stresses and stress divergence terms. Johansson et al. (2003) argued that higher order velocity moments of an axi-symmetric wake can be shown to develop downstream over large distances and continue to contain information of the near wake. It remains unclear if this conclusion can be applied to a utility scale wind turbine wake due to the mismatch in Reynolds-number.

Using the normalized coordinates $\tilde{y} = y/D$ and $\tilde{z} = (z - z_H)/D$, along with the four normalized parameters $\Delta \tilde{U}_{\text{max}} = \Delta U_{\text{max}}/U_H$, $\tilde{\sigma} = \sigma/D$, $\tilde{z}_H = z_H/D$ and $\tilde{z}_0 = z_H/z_0$, the streamwise velocity, shear stresses and stress divergence terms can

be written in a dimensionless form:

$$g\left(\tilde{y},\tilde{z}\right) \equiv \frac{\Delta U_{\text{wake}}}{U_H} = \Delta\tilde{U}_{\max} \cdot e^{-\left(\tilde{y}^2+\tilde{z}^2\right)/2\tilde{\sigma}^2},\tag{9}$$

$$f\left(\tilde{y},\tilde{z}\right) \equiv \frac{U(y,z)-U_H}{U_H} = \frac{\ln\left(\tilde{z}/\tilde{z}_H+1\right)}{\ln\left(\tilde{z}_0\right)} - g\left(\tilde{y},\tilde{z}\right),\tag{10}$$

$$f'_\alpha\left(\tilde{y},\tilde{z}\right) \equiv \frac{-e^{1/2}\sigma}{u_*\kappa z_H U_H}\left(\overline{u'u'_\alpha} - \overline{u'u'_{\alpha\text{in}}}\right) = \left(1+\frac{\tilde{z}}{\tilde{z}_H}\right)\left(\frac{\tilde{y}}{\tilde{\sigma}}\delta_{2\alpha}+\frac{\tilde{z}}{\tilde{\sigma}}\delta_{3\alpha}\right)e^{1/2}g\left(\tilde{y},\tilde{z}\right),\tag{11}$$

$$f''_\alpha\left(\tilde{y},\tilde{z}\right) \equiv -\frac{\sigma^2}{u_*\kappa z_H U_H}\frac{\partial}{\partial x_\alpha}\left(\overline{u'u'_\alpha}\right) = \left(1+\frac{\tilde{z}}{\tilde{z}_H}\right)\left(1-\frac{\tilde{y}^2}{\tilde{\sigma}^2}\delta_{2\alpha}+\left[\frac{\tilde{z}}{\tilde{z}+\tilde{z}_H}-\frac{\tilde{z}^2}{\tilde{\sigma}^2}\right]\delta_{3\alpha}\right)g\left(\tilde{y},\tilde{z}\right),\tag{12}$$

with $\overline{u'u'_{\alpha\text{in}}} = -\delta_{3\alpha}u_*^2$ as the inflow shear stress and $\delta_{i\alpha}$ as the Kronecker delta. In addition, the Greek index $\alpha$ is used to show that summation is not performed over the indices, and will be used throughout this article.

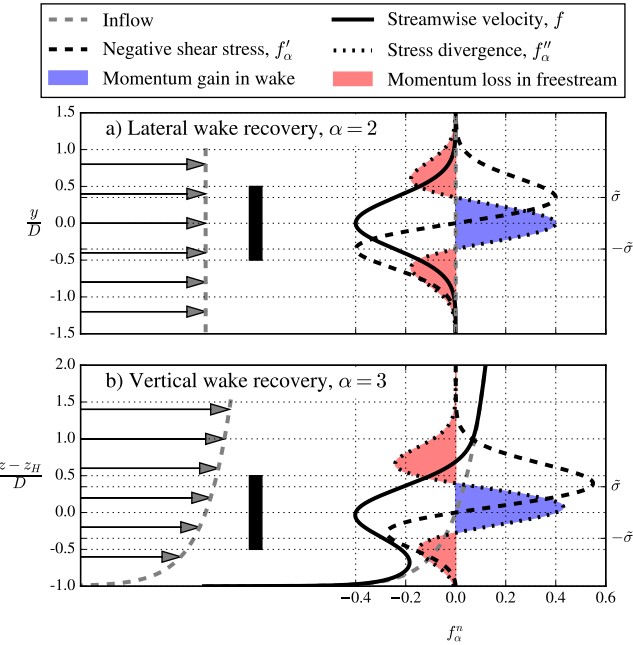

**Figure 1.** Gaussian velocity wake deficit, negative shear stress and stress divergence using $\Delta\tilde{U}_{\max} = 0.4$, $\tilde{\sigma} = 0.35$, $\tilde{z}_H = 1$, $\tilde{z}_0 = 10^4$. a) Lateral wake recovery at hub height. b) Vertical wake recovery at the rotor center. Black filled rectangle indicates the rotor area.

The results of the far wake model are depicted in Fig. 1, in terms of normalized streamwise velocity, $f$, normalized Reynolds shear stresses, $f'_\alpha$ and normalized Reynolds stress divergence, $f''_\alpha$, as function of the lateral ($\alpha = 2$) and vertical distance ($\alpha = 3$). Note that the prime indicates the derivative of $f$ times a normalization factor, i.e., $f'_\alpha \neq \partial f/\partial x_\alpha$. The results in Fig. 1 are made with $\Delta\tilde{U}_{\max} = 0.4$, which could reflect a certain downstream distance, although the overall behavior is not influenced by $\Delta\tilde{U}_{\max}$. Figure 1a shows the first and second derivatives of the wake deficit that represent the negative shear stress $-\overline{u'v'}$ and stress divergence $-\partial\overline{u'v'}/\partial y$. The stress divergence is negative at the wake edges and positive at the wake center, which shows how momentum outside the wake is transported to the wake center; this is the main mechanism for (far) wake recovery.

A similar observation can be made in the vertical wake recovery depicted in Fig. 1b; however, more momentum from above is transported to the center with respect to the bottom due the eddy viscosity of the logarithmic inflow that increases linearly with height. It can easily be shown that the integral of the negative stress divergence (depicted by the red areas in Fig. 1a) is equal to the integral of the positive stress divergence (depicted by the blue area in Fig. 1a). This must hold because stress divergence is momentum transport and should not result in a loss or gain of total momentum. The same balance of stress divergence is present in the vertical wake recovery depicted in Fig. 1b. The amount of lateral $U$-momentum transfer, $M_{\mathrm{lateral}}$, and vertical $U$-momentum transfer at bottom, $M_{\mathrm{vertical,b}}$, and top of the wake, $M_{\mathrm{vertical,t}}$, can be quantified by the bottom and top peaks of $\overline{u'v'}$ and $\overline{u'w'}$, respectively, since we can write:

$$M_{\mathrm{lateral}} = \int_{-\infty}^{y_-} \frac{\partial \overline{u'v'}}{\partial y} dy + \int_{y_+}^{\infty} \frac{\partial \overline{u'v'}}{\partial y} dy = \overline{u'v'}|_{y_-} - \overline{u'v'}|_{y_+} = 2e^{-1/2}\frac{\nu_T}{\sigma}\Delta U_{\mathrm{max}} \tag{13}$$

$$M_{\mathrm{vertical,b}} = \int_{0}^{z_-} \frac{\partial \overline{u'w'}}{\partial z} dz = \overline{u'w'}|_{z_-} + u_*^2, \qquad M_{\mathrm{vertical,t}} = \int_{z_+}^{\infty} \frac{\partial \overline{u'w'}}{\partial z} dz = -\overline{u'w'}|_{z_+} - u_*^2 \tag{14}$$

where $y_-$ and $y_+$ are the lateral locations of the peaks in $\overline{u'v'}$, and $z_-$ and $z_+$ are the vertical locations of the bottom and the top peaks in $\overline{u'w'}$, respectively. For the analytic model, we have $y_- = -\sigma$, $y_+ = \sigma$, and $z_-$ and $z_+$ are solutions of the cubic equation $2(z/z_H) - (z/z_H)[(z - z_H)/\sigma]^2 = 1$, see Eqs. (7)-(8). The analytical model predicts that the momentum transfer from above is larger then the momentum transfer from below, as shown by the peak values of $\overline{u'w'}$ in Fig. 1, i.e., $M_{\mathrm{vertical,t}} > M_{\mathrm{vertical,b}}$. Calaf et al. (2010) performed a related analysis on a large wind farm LES data set by integrating the horizontally averaged vertical kinetic energy flux over the rotor area; the obtained result was shown to be in order of the power extracted by wind turbines.

The fact that wake recovery requires a change of shear also becomes clear when considering a homogeneous shear flow (Pope, 2000), a temperature diffusion equation or the Ekman spiral; these examples are further discussed in Appendix B.

## 3  Wake recovery in a large-eddy simulation

The wake recovery in terms of the stress divergence of $\overline{u'u'_\alpha}$ is post processed from an LES of a single wind turbine wake in a neutral pressure-driven atmospheric boundary layer. The LES is the same as used by Hornshøj-Møller et al. (2021); numerical details can be found in Abkar and Porté-Agel (2015). The wind turbine represents a Vestas V80 wind turbine that has a rotor diameter and hub height of 80 and 70 m, respectively. The wind turbine forces are modeled as an AD and has an effective thrust coefficient of 0.77. The inflow wind speed and total turbulence intensity at hub height are 8.0 ms$^{-1}$ and 5.7%, respectively.

Figure 2 shows the normal, lateral and vertical stress divergence that contribute to the streamwise momentum equation at hub height and at a vertical plane through the rotor center. The normal stress divergence has the largest (negative) values in the near wake (Figs. 2a) and d)) but is about five times smaller than the shear stress divergence based on volumetric integrals of

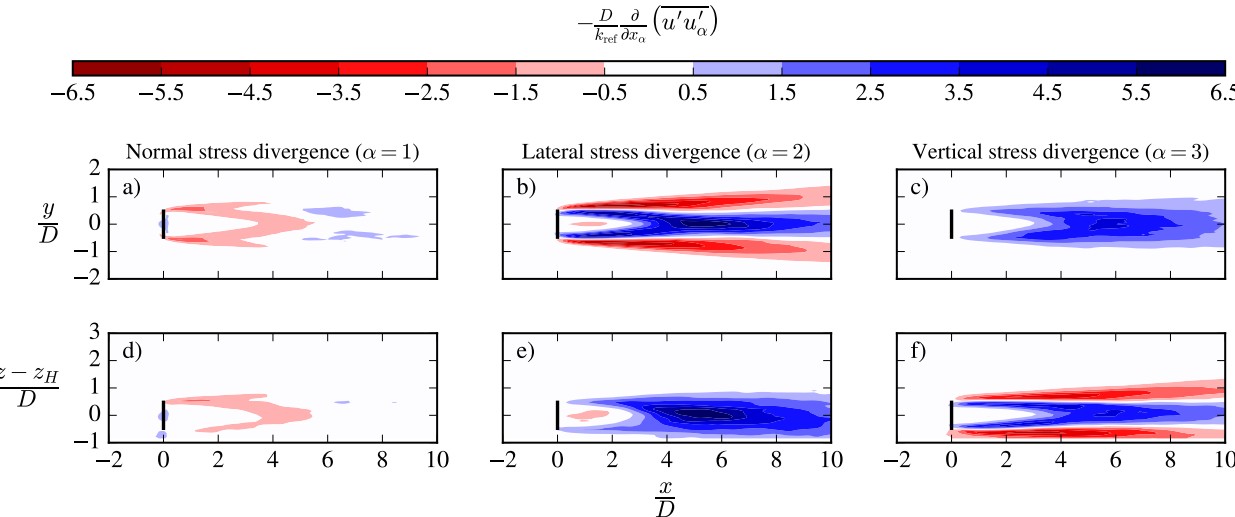

**Figure 2.** Wake recovery in terms of stress divergence from an LES single wake simulation. a)-c) Lateral wake recovery at hub height. d)-f) Vertical wake recovery at $y = 0$.

the three streamwise stress divergence terms:

$$M_\alpha = \int\limits_V \left| \frac{\partial \overline{u'u'_\alpha}}{\partial x_\alpha} \right| dV \tag{15}$$

where $V$ is a box around the wind turbine located at $(x, y, z) = (0, 0, z_H)$ with dimensions $-2 \le x/D \le 20$, $-2 \le y/D \le 2$ and $0 \le (z - z_H)/D \le 3.125$. We obtain $M_2/M_1 = 5.5$, $M_3/M_1 = 5.1$. Hence, the LES data shows that it is mainly the shear stress divergence that leads to wake recovery by bringing momentum from the freestream into the wake center (best visible in Figs. 2b) and f)). The shear stress divergence represents wake meandering and turbulent cross diffusion, which is slightly larger in the lateral direction compared to the vertical direction due to the ground ($M_3/M_2 = 0.93$), although the atmospheric conditions and presence of neighboring wind turbines may influence the dominant direction of wake recovery. The normal stress divergence represents the streamwise back and forth movement of the wake and streamwise turbulent diffusion, which is much less compared to the lateral and vertical wake recovery.

The LES-based lateral and vertical wake recovery is depicted in Fig. 3, at three different downstream locations: $x/D = 2.5$, 5 and 7.5. Results of the streamwise velocity, negative shear stress and shear stress divergence are shown; they are normalized in the same way as performed for the analytical far wake model as defined by Eqs. (10)-(12) and depicted in Fig. 3. The normalized standard deviation, $\tilde{\sigma}$, used for normalization of the shear stress and its divergence is obtained by a Gaussian fit with the velocity deficit at each downstream location ($\tilde{\sigma} = 0.37, 0.39, 0.45$ and $\tilde{\sigma} = 0.36, 0.37, 0.44$ for the lateral and vertical wake recovery at $x/D = 2.5$, 5 and 7.5, respectively). Furthermore, we have used $u_* = 0.333$ ms$^{-1}$ and $U_H = 8.0$ ms$^{-1}$. A similar behavior of the LES-derived velocity deficit, shear stresses and the stress divergence is obtained at $x/D = 5$ and 7.5 compared to the analytic far wake model, as depicted in Fig. 1. As discussed previously, the lateral wake recovery is slightly

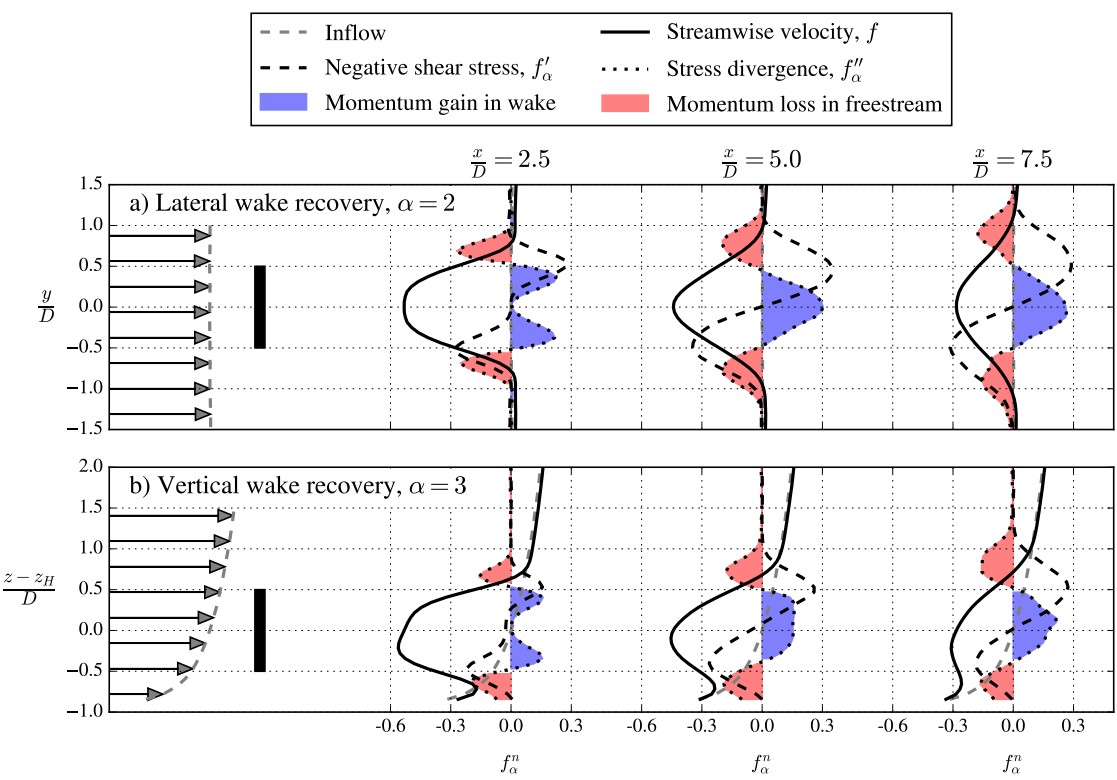

**Figure 3.** Normalized profiles of streamwise velocity, negative shear stress and shear stress divergence from a single wake LES.

stronger than the vertical wake recovery; the latter is stronger at the top of the wake with respect to the bottom of the wake, as expected and predicted by the simple far wake model. The results in the near wake ($x/D = 2.5$) are different in LES, as also

expected, where the wake center momentum gain has a double bell shape due to the more complex shear stress profile.

The fact that the normal stress divergence is an order of magnitude smaller than the combination of lateral and vertical shear stress divergence indicates that RANS turbulence model closures do not need to model the anistropy of the normal Reynolds stresses if the velocity deficit is the only quantity of interest. This implies that one could rely on the isoptropic hypothesis of Boussinesq (1897), as long as the turbulence model is able to predict correct shear stresses and give realizable Reynolds

stresses – for example by using a flow dependent eddy viscosity coefficient that limits the turbulence length scale (van der Laan and Andersen, 2018). Whether this also applies in stratified atmospheric conditions is a subject for further studies.

## 4   Conclusions

The main mechanisms of wake recovery are explained by the stress divergence, considering both a Gaussian-based analytical far-wake model and LES of a single wind turbine in neutral atmospheric conditions. The LES data shows that the divergence

of the lateral and vertical shear stresses combined are an order of magnitude larger than the divergence of the normal stresses;

i.e., $\partial \overline{u'v'}/\partial y$ and $\partial \overline{u'w'}/\partial z$—and not simply 'shear'—are the main contributors to wake recovery. The analytical model qualitatively captures the behavior of the stress divergence observed in the far wake of the LES results, which shows that the *second* derivatives $\partial^2 U/\partial y^2$ and $\partial^2 U/\partial z^2$ induce wake recovery. This also indicates that RANS turbulence model closures only need to be able to model the shear stresses accurately, if the velocity deficit is the sole quantity of interest.

## 160  Appendix A: Influence of non-uniform eddy viscosity on a single wind turbine wake in RANS

The simple far wake model of Sect. 2 has been derived using a constant eddy viscosity, while a real wind turbine far wake is expected to result in non-uniform eddy viscosity. In order to quantify the impact of this assumption, two RANS simulations of single wind turbine are employed based on Case 5 of van der Laan et al. (2015b). In this previous work, the turbulence was modeled by the $k$-$\varepsilon$-$f_P$ model (van der Laan et al., 2015b), the wind turbine forces were represented by an actuator

disk ($C_T = 0.79$) and the inflow was a logarithmic surface layer (using a turbulence intensity at hub height of 4%). The numerical setup of the present study is the same as performed in previous work (van der Laan et al., 2015b) with a number of modifications. First of all, a longer and wider refined domain around the wind turbine is used ($y = \pm 4D$ in the lateral direction, and $35D$ downstream in streamwise direction) in order to resolve the wake up to $x/D = 30$. In addition, a uniform inflow is applied and the ground (modeled as a rough wall boundary condition) is removed by using the same cell distribution and

periodic boundary conditions for the vertical coordinate as is used for the lateral coordinate. Finally, the inflow eddy viscosity is uniform and set equal to the hub height eddy viscosity of a logarithmic inflow; it is maintained by using ambient $k$ and $\varepsilon$ source terms in the $k$ and $\varepsilon$ transport equations similar to van der Laan et al. (2015a). The removal of the ground decreases the velocity deficit because the wake recovery is enhanced; however, the latter is necessary for modeling a uniform inflow eddy viscosity. One RANS simulation is employed with this setup and represents a variable eddy viscosity case. A second RANS simulation

is performed by prescribing the eddy viscosity equal to the inflow in the entire domain, which means that the wind turbine does not change the eddy viscosity in the wake. This represents a constant eddy viscosity case, as assumed for the simple far wake model of Sect. 2. The second RANS simulation is equivalent to a laminar wind turbine wake simulation without a turbulence model by setting a low Reynolds number of $Re = DU_H/\nu_T$. Figure A1 depicts the velocity deficit (Figs. A1a-c) and eddy viscosity (Figs. A1d-f) at hub height of both RANS simulations for three downstream distances ($x/D = 7.5, 15, 30$).

As expected, a larger velocity deficit for the constant eddy viscosity RANS simulation is obtained compared to the velocity deficit of RANS simulation with a variable eddy viscosity. However, our focus of Fig. A1 are the Gaussian functions that have been fitted to the velocity deficits. Figure A1a shows that velocity deficit of the constant eddy viscosity simulation compares better with a Gaussian function with respect to the velocity deficit of the variable eddy viscosity simulation, although the differences of latter are small (mainly visible at the wake center (y=0) and at the wake edges). This deviation reduces further

downstream as shown in Figs A1b and c. The results in Figs. A1a-c are also depicted in Fig. A2 in a form where the self-similarity of the velocity deficit of the two RANS simulations can be compared. Here, $y_{1/2}$ represents the half wake width of the velocity deficit, which is the lateral location at which half the velocity deficit is obtained. We obtain similar trends as found by Bastankhah and Porté-Agel (2014); Cafiero et al. (2020), where results of LES and wind tunnel measurements of a

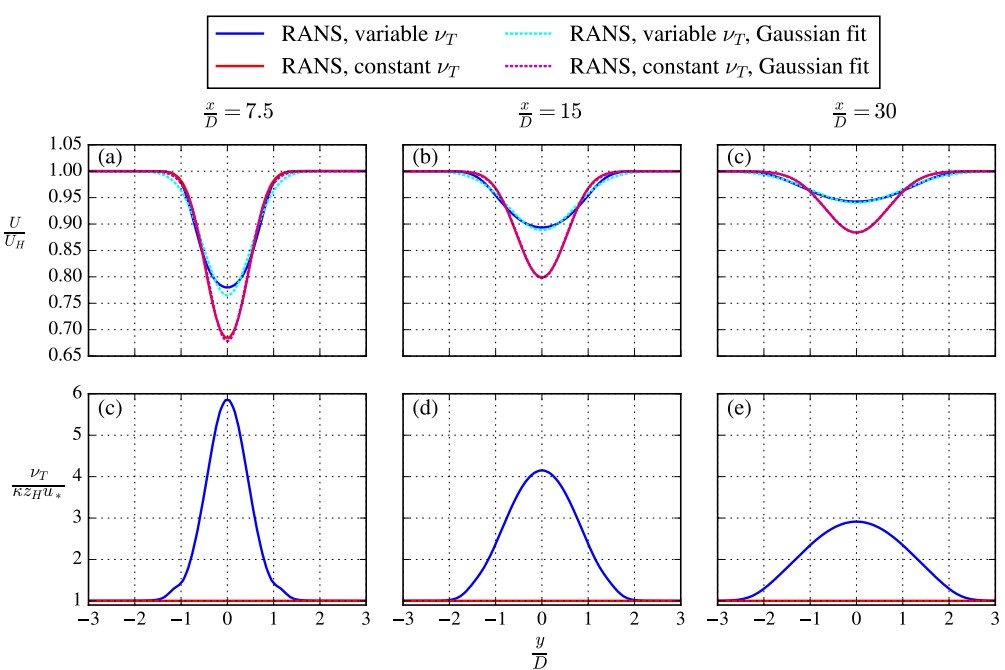

**Figure A1.** Velocity deficit (a-c) and eddy viscosity (d-f) of single wind turbine simulated in RANS.

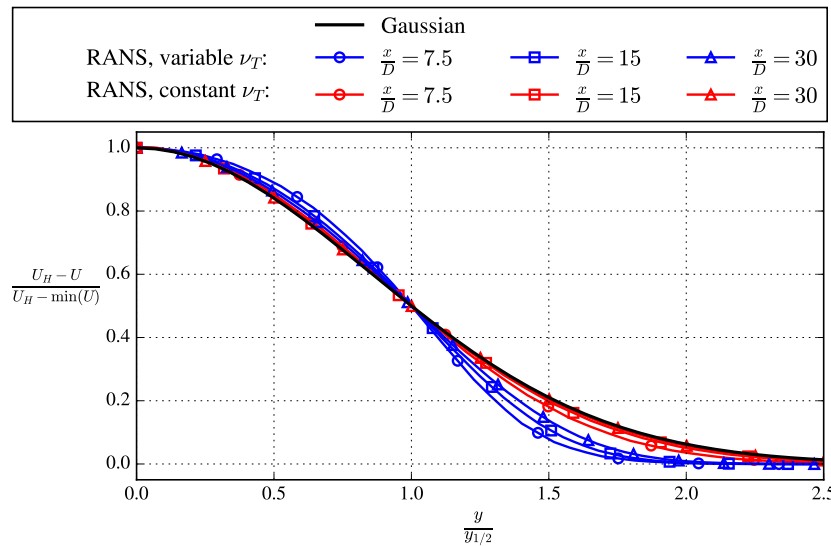

**Figure A2.** Self-similarity of the velocity deficit of a single wind turbine wake in RANS.

wake also indicated smaller tails of the far wake velocity deficit compared to a Gaussian function. The RANS results presented
here suggest that a variable eddy viscosity could be an explanation similar to Cafiero et al. (2020). One other explanation why

results of LES and measurements predict smaller tails could be related to the amount of data necessary to obtain converged statistics, which becomes more demanding at the wake edges further downstream. It should be noted that the deviation of the RANS simulated velocity deficits with the Gaussian function using a variable eddy viscosity, as presented in Figs. A1 and A2, can be dependent on the chosen turbulence model. The employed turbulence model was developed to obtain accurate velocity deficits compared to LES (van der Laan et al., 2015b) but does not guarantee accurate results for the eddy viscosity.

## Appendix B: Other examples of stress divergence

Section 2 showed how the stress divergence, i.e. the gradient of the shear within a Boussinesq/eddy-viscosity framework, 'recovers' a wind turbine wake. This becomes more clear when considering a hypothetical flow that includes a constant shear and a constant eddy viscosity in space, without a pressure gradient, since in this case the right hand side of the momentum equation will be zero and the shear will not recover to a uniform flow. This flow is also known as a homogeneous shear flow (Pope, 2000) and it is often used to test turbulence model equations without the influence of an active momentum equation. A homogeneous shear flow case is analogous to modeling an initial constant temperature gradient, $dT/dz$, with a simple heat diffusion equation using bottom, $z_1$ and top, $z_2$ boundary conditions that set a fixed low and high temperature values, $T_1$, and $T_2$, respectively, since the heat diffusion equation would also be in balance in this case:

$$\frac{\partial T}{\partial t} = k\frac{\partial^2 T}{\partial z^2} = 0, \quad T_1 \equiv T|_{z=z_1}, \quad T_2 \equiv T|_{z=z_2}, \quad T|_{t=0} = \frac{T_2 - T_1}{z_2 - z_1}(z - z_1) + T_1 \tag{B1}$$

with $T(t,z)$ as the temperature as function of time $t$ and spatial variable $z$, and $k$ as the diffusivity constant.

Another well-known example where the role of stress divergence becomes clear is the Ekman spiral (Ekman, 1905), which is an analytic solution of the Ekman equations (often written in complex form) that describe a boundary layer profile including Coriolis forces using a constant eddy viscosity:

$$\nu_T \frac{d^2\hat{W}}{dz^2} = if_c W, \quad \hat{W}(z=0) = -U_G - iV_G, \quad \hat{W}(z=\infty) = 0; \tag{B2}$$

the complex velocity vector is $\hat{W} = U - U_G + i(V - V_G)$, where $i \equiv \sqrt{-1}$. $U_G$ and $V_G$ are the streamwise and lateral geostrophic wind speed components, respectively, and $f_c$ is the Coriolis parameter. The well-known Ekman solution can then be written as

$$\hat{W} = -(U_G + iV_G)e^{[-(i+1)\gamma z]}, \tag{B3}$$

with $\gamma = \sqrt{f_c/(2\nu_T)}$. If the wind direction is set to be zero at $z=0$ by using $U_G = -V_G$ and a positive $f_c$ then the integral of the streamwise velocity profile minus the (constant) streamwise geostrophic wind speed, $U_G$, is zero (Wyngaard, 2010):

$$\int\limits_{\xi=0}^{\xi=\infty} \frac{U - U_G}{U_G} d\xi = \int\limits_{\xi=0}^{\xi=\infty} e^{-\xi}(\sin\xi - \cos\xi)\, d\xi = 0, \tag{B4}$$

with $\xi = \gamma z$. This integral is depicted in Fig. B1 and is similar to the integral of stress divergence shown in Fig. 1. The horizontal dashes lines in Fig. B1 depict transitions between momentum loss and gain located at $\gamma z = \pi/4 + n\pi$, with $n$ as a positive integer.

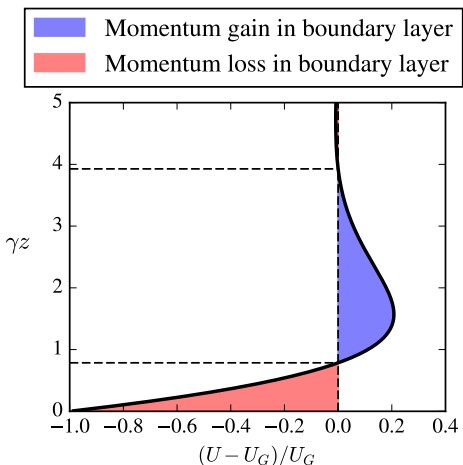

**Figure B1.** Stress divergence in an Ekman spiral.

*Author contributions.* MPVDL has drafted the article, produced the figures and performed the RANS simulations. MB pointed out the stress divergence balance (visualized with filled colors) and has post processed the LES data set. MK suggested the generic connection to the Ekman spiral from Wyngaard (2010) and general stress divergence mechanism. All authors contributed to the methodology and finalization of the paper.

*Competing interests.* The authors declare that they have no conflict of interest.

*Acknowledgements.* We would like to thank Mahdi Abkar for supplying the LES data.

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
