# Peer review of "Brief communication: A clarification of wake recovery mechanisms"

_Wind Energy Science, 2022_

## Referee Comment (RC1)

*Brief communication: A clarification of wake recovery mechanisms.*

*authors:*
Maarten Paul van der Laan
Mads Baungaard
and Mark Kelly1

**Summary:**

The manuscript entitled "Brief communication: A clarification of wake recovery mechanisms" endeavors to describe the recovery mechanisms of a wind turbine wake in terms of fundamental flow quantities. The manuscript points toward several commonly accepted relationships between mean sheer gradients, turbulent stresses, and the recovery of momentum deficits, relying heavily on simplified wake models and classical hypotheses. The primary argument of the authors is that "the gradient of the shear is responsible for wake recovery, and not the shear or Reynolds stresses themselves." While this manuscript aims to clarify that is often missed in the wind turbine wake research community, the momentum recovery is more cleanly described by the flux and production terms in the mean kinetic energy transport equation. This topic has been explored to some depth going as far back as a few decades. The authors would be well served to survey the literature in more depth.

**Comments:**

- Viewed from the perspective of the mean and turbulent kinetic energy transport equations, the divergence of the turbulent stresses are found in a flux term of the form, $\frac{\partial \overline{u_i u_j} U_i}{\partial x_i}$. Expanding with the chain rule offers the very quantities discussed at length in the manuscript, although from a fundamental starting point, rather than a phenomenological one.
- The simplified description of wake contributions relies heavily on many assumptions and hypotheses including:
  - neglecting $\partial / \partial x$ terms
  - zero pressure gradient in the far wake
  - constant eddy viscosity in the wake
  - isotropic Boussinesq hypothesis
  - no buoyancy terms or thermally-driven turbulence
  - etc.

  Some of these assumptions are questionable in real-world scenarios, and should be addressed more clearly in the manuscript. Are there cases where the assumptions do not hold that would change the balance of wake recovery mechanisms (e.g., strongly convective atmospheric boundary layers)?
- Under what conditions would a non-uniform eddy viscosity change the balance of wake recovery mechanism?

- line 45, "... $z_H$ is the wind turbine hub height and $\sigma$ is the standard deviation..." of what, exactly? What are the units? It would help readers and researchers new to the field to explain the details of the model, and why it is a good choice.
- Lines 78–80 relate the magnitude of the momentum transport to the local extrema of the divergence of the shear stresses. This builds on a large body of work that is not cited in the manuscript (see references below, to start).
- The authors point to two cases that would ostensibly illustrate the main argument (namely, the homogeneous shear flow and the Ekman spiral), but do not actually show support for their thesis, either mathematically or graphically. If these cases support the main argument, the paper would be greatly strengthened with more complete demonstrations.

**References**

[1]   Sten Frandsen. "On the wind speed reduction in the center of large clusters of wind turbines". In: *Journal of Wind Engineering and Industrial Aerodynamics* 39.1-3 (1992), pp. 251–265.

[2]   Raúl Bayoán Cal et al. "Experimental study of the horizontally averaged flow structure in a model wind-turbine array boundary layer". In: *Journal of renewable and sustainable energy* 2.1 (2010), p. 013106.

[3]   Marc Calaf, Charles Meneveau, and Johan Meyers. "Large eddy simulation study of fully developed wind-turbine array boundary layers". In: *Physics of fluids* 22.1 (2010), p. 015110.

[4]   Nicholas Hamilton et al. "Statistical analysis of kinetic energy entrainment in a model wind turbine array boundary layer". In: *Journal of renewable and sustainable energy* 4.6 (2012), p. 063105.

[5]   Johan Meyers and Charles Meneveau. "Optimal turbine spacing in fully developed wind farm boundary layers". In: *Wind energy* 15.2 (2012), pp. 305–317.

[6]   Nicholas Hamilton, Murat Tutkun, and Raúl Bayoán Cal. "Wind turbine boundary layer arrays for Cartesian and staggered configurations: Part II, low-dimensional representations via the proper orthogonal decomposition". In: *Wind Energy* 18.2 (2015), pp. 297–315.

[7]   Naseem Ali et al. "Turbulence kinetic energy budget and conditional sampling of momentum, scalar, and intermittency fluxes in thermally stratified wind farms". In: *Journal of Turbulence* 20.1 (2019), pp. 32–63.

---

## Author Response (AR1)

**Reply to reviewers**

January 18, 2023

We would like to thank the two reviewers for their detailed feedback and suggestions to improve the article. In the next sections, the reviewers comments are copied and answered per comment (blue color). An additional document is provided that highlights all modifications with respect to the initial submitted version.

**Reviewer 1**

The manuscript entitled "Brief communication: A clarification of wake recovery mechanisms" endeavors to describe the recovery mechanisms of a wind turbine wake in terms of fundamental flow quantities. The manuscript points toward several commonly accepted relationships between mean sheer gradients, turbulent stresses, and the recovery of momentum deficits, relying heavily on simplified wake models and classical hypotheses. The primary argument of the authors is that "the gradient of the shear is responsible for wake recovery, and not the shear or Reynolds stresses themselves." While this manuscript aims to clarify that is often missed in the wind turbine wake research community, the momentum recovery is more cleanly described by the flux and production terms in the mean kinetic energy transport equation. This topic has been explored to some depth going as far back as a few decades. The authors would be well served to survey the literature in more depth.

1. Viewed from the perspective of the mean and turbulent kinetic energy transport equations, the divergence of the turbulent stresses are found in a flux term of the form, $\partial \overline{u_i' u_j'} U_i / \partial x_i$. Expanding with the chain rule offers the very quantities discussed at length in the manuscript, although from a fundamental starting point, rather than a phenomenological one.

    Thank you for those references. We are aware of the kinetic energy approach and we had referred to Newman et al. (2014) and Andersen et al. (2017). We have avoided a long literature study as a brief communication article only allows a total of 20 references. We do understand that including more literature can strengthen the article. We have made an overview of the main results and conclusions in those articles:

    - Frandsen (1992): Derived an analytic model for the wind farm wake reduction of an infinite wind farm and effective wind farm roughness using balances of momentum and vertical energy transport.

    - Cal et al. (2010): Performed wind tunnel measurements of a small scale wind farm with the aim of studying the horizontally averaged flow of a wind farm. The main results are horizontal averaged profiles of several terms from the kinetic energy equation. The authors conclude that the shear-stress is the main contribution to kinetic energy recovery in a wind farm.

    - Calaf et al. (2010): Performed atmospheric large eddy simulations of a large wind farm with periodic lateral boundary conditions to simulate a fully developed wind farm flow. They derive effective wind farm roughnesses and conclude that the vertical fluxes of kinetic energy are the main contributors to the kinetic energy recovery in a large wind farm.

    - Meyers and Meneveau (2012): Employed the LES data from Calaf et al. (2010) to determine optimal wind turbine spacing for energy efficiency.

- Hamilton et al. (2012): Analyzed kinetic energy entrainment of a small scale wind farm, measured in a wind tunnel. The main results are vertical profiles of horizontally averaged quantities as wind speed, Reynolds-streses and vertical fluxes of kinetic energy. A quandrant decomposition is performed and indicates that sweep motions dominate energy entrainment from above the wake, while ejection events dominate below and inside the wake.

- Hamilton et al. (2014): Performed proper orthogonal decomposition to reconstruct Reynolds-stresses of a small scale single wind turbine wake, measured in a wind tunnel.

- Ali et al. (2019): Performed turbulent kinetic energy budget analysis with large eddy simulations of a large wind farm subjected to a diurnal cycle, in order to include effects of atmospheric stability. The main results are vertical profiles of kinetic energy budget terms and results of a quadrant analysis.

While the provided references discuss relevant topics for wind turbines wakes and wind farm flows they are not all required for the present brief communication study. We have decided to add Cal et al. (2010) and Calaf et al. (2010) to the introduction as they are the most relevant to include, i.e.:

*"Other authors have analyzed the mean kinetic energy budget of a wind farm using wind tunnel measurements (Cal et al., 2010; Newman et al., 2014) and large-eddy simulations (LES, e.g. Calaf et al., 2010; Andersen et al., 2017); they concluded that the vertical shear stress component of the Reynolds stress is the main driver behind energy transport of the freestream into the wake."*

We do find that the references provided by the reviewer never mention stress divergence, but instead focus on the mean kinetic energy equation, which further motivates the main goal of the present article. The mean kinetic energy equation can be useful as most of the above references show, but we find it more intuitive and simpler to directly plot the stress divergence in order to visualize the wake recovery and to obtain the most simple model that we can think of. If we had chosen to make a simple model based on the kinetic energy equation then the result would have been more complicated. Note that we do not investigate terms as $\partial \overline{u_i' u_j'} U_i / \partial x_i$ because we have not multiplied the momentum equation by the mean streamwise velocity. Instead, we simply look at the momentum terms directly and integrate the stress divergence over a line (simple model) or volume (large eddy simulation data) in order to quantify the wake recovery.

2. The simplified description of wake contributions relies heavily on many assumptions and hypotheses including:

- neglecting $\partial / \partial x$ terms
- zero pressure gradient in the far wake
- constant eddy viscosity in the wake
- isotropic Boussinesq hypothesis
- no buoyancy terms or thermally-driven turbulence
- – etc.

Some of these assumptions are questionable in real-world scenarios, and should be addressed more clearly in the manuscript. Are there cases where the assumptions do not hold that would change the balance of wake recovery mechanisms (e.g., strongly convective atmospheric boundary layers)?

It is correct that these assumptions could be violated in real world scenarios. Although assuming fully developed conditions for the far wake (i.e neglecting $\partial / \partial x$ and a zero pressure gradient), a constant eddy viscosity and isotropic turbulence do not seem to change the overall wake recovery mechanisms, as shown by the large-eddy simulation results for neutral conditions from Figs. 2 and 3. We suspect that the effect atmospheric stability can be significant and may alter the magnitude of the lateral and vertical wake recovery. For example, the vertical momentum wake recovery is expected to be lower for stable atmospheric conditions compared to neutral stratification. We have added/ modified the following in Sect. 1:

*"Here, we have assumed that the eddy viscosity is unaffected by the wake and equal to the logarithmic*

*inflow: $\nu_T = u_* \kappa z$ by assuming a neutral atmospheric surface layer to be valid. This is a strong assumption and does not hold for non-neutral atmospheric conditions and for tall wind turbines that may operate beyond the surface layer."*

3. Under what conditions would a non-uniform eddy viscosity change the balance of wake recovery mechanism?

   *We suspect that it is not very important, as large-eddy simulation results (Bastankhah and Porté-Agel, 2014; Xie and Archer, 2015) of a far wake velocity deficit (including a non-uniform eddy viscosity) match relatively well with a Gaussian profile. The latter is also an analytic result of assuming constant eddy viscosity (Pope, 2000) and thus provides an indirect proof. However, the second reviewer also had comments regarding this point and we have added a discussion in Sect. 1 and an Appendix A regarding uniform vs non-uniform eddy viscosity profiles, which shows that the self-similarity profile of the far wake deficit is not a Gaussian (Fig. A2). However, this deviation is small when looking at the difference in absolute numbers relevant to wind energy application and the difference decreases with downstream distance (Fig. A1).*

4. line 45, "... $z_H$ is the wind turbine hub height and $\sigma$ is the standard deviation..." of what, exactly? What are the units? It would help readers and researchers new to the field to explain the details of the model, and why it is a good choice.

   *We think that the hub height is trivial for a reader of Wind Energy Science. We do understand the confusion with the standard deviation and we have modified line 45 as: "...$z_H$ is the wind turbine hub height and $\sigma$ is the standard deviation of the Gaussian wake profile."*

5. Lines 78–80 relate the magnitude of the momentum transport to the local extrema of the divergence of the shear stresses. This builds on a large body of work that is not cited in the manuscript (see references below, to start).

   *While none of those references mention stress divergence, Calaf et al. (2010) did indeed perform a similar integral analysis. We have added the following in Sect. 1:*
   *"Calaf et al. (2010) performed a related analysis on a large wind farm LES data set by integrating the horizontally averaged vertical kinetic energy flux over the rotor area; the obtained result was shown to be in order of the power extracted by wind turbines."*

6. The authors point to two cases that would ostensibly illustrate the main argument (namely, the homogeneous shear flow and the Ekman spiral), but do not actually show support for their thesis, either mathematically or graphically. If these cases support the main argument, the paper would be greatly strengthened with more complete demonstrations.

   *You are right about this and we agree that more clarification is necessary, which we have added in an Appendix (Appendix B):*
   *Section 1 showed how the stress divergence, i.e. the gradient of the shear within a Boussinesq/eddy-viscosity framework, 'recovers' a wind turbine wake. This becomes more clear when considering a hypothetical flow that includes a constant shear and a constant eddy viscosity in space, without a pressure gradient, since in this case the right hand side of the momentum equation will be zero and the shear will not recover to a uniform flow. This flow is also known as a homogeneous shear flow (Pope, 2000) and it is often used to test turbulence model equations without the influence of an active momentum equation. A homogeneous shear flow case is analogous to modeling an initial constant temperature gradient, $dT/dz$, with a simple heat diffusion equation using bottom, $z_1$ and top, $z_2$ boundary conditions that set a fixed low and high temperature values, $T_1$, and $T_2$, respectively, since the heat diffusion equation would also be in balance in this case:*

$$\frac{\partial T}{\partial t} = k\frac{\partial^2 T}{\partial z^2} = 0, \quad T_1 \equiv T|_{z=z_1}, \quad T_2 \equiv T|_{z=z_2}, \quad T|_{t=0} = \frac{T_2 - T_1}{z_2 - z_1}(z - z_1) + T_1 \qquad (1)$$

with $T(t, z)$ as the temperature as function of time $t$ and spatial variable $z$, and $k$ as the diffusivity constant.

Another well-known example where the role of stress divergence becomes clear is the Ekman spiral (Ekman, 1905), which is an analytic solution of the Ekman equations (often written in complex form) that describe a boundary layer profile including Coriolis forces using a constant eddy viscosity:

$$\nu_T \frac{d^2\hat{W}}{dz^2} = if_c W, \quad \hat{W}(z = 0) = -U_G - iV_G, \quad \hat{W}(z = \infty) = 0; \tag{2}$$

the complex velocity vector is $\hat{W} = U - U_G + i(V - V_G)$, where $i \equiv \sqrt{-1}$. $U_G$ and $V_G$ are the streamwise and lateral geostrophic wind speed components, respectively, and $f_c$ is the Coriolis parameter. The well-known Ekman solution can then be written as

$$\hat{W} = -(U_G + iV_G)e^{[-(i+1)\gamma z]}, \tag{3}$$

with $\gamma = \sqrt{f_c/(2\nu_T)}$. If the wind direction is set to be zero at $z = 0$ by using $U_G = -V_G$ and a positive $f_c$ then the integral of the streamwise velocity profile minus the (constant) streamwise geostrophic wind speed, $U_G$, is zero (Wyngaard, 2010):

$$\int_{\xi=0}^{\xi=\infty} \frac{U - U_G}{U_G} d\xi = \int_{\xi=0}^{\xi=\infty} e^{-\xi}(\sin \xi - \cos \xi) d\xi = 0, \tag{4}$$

with $\xi = \gamma z$. This integral is depicted in Fig. B1 and is similar to the integral of stress divergence shown in Fig. 1. The horizontal dashed lines in Fig. B1 depict transitions between momentum loss and gain located at $\gamma z = \pi/4 + n\pi$, with $n$ as a positive integer.

[Figure]

Figure 1: Stress divergence in an Ekman spiral.

**Reviewer 2**

The proposed manuscript discusses wake recovery mechanisms for wind turbines. An analytical model, assuming constant eddy viscosity within the Boussinesq hypothesis, is developed to assess the momentum transfer from the divergence of Reynolds shear stresses. Furthermore, large eddy simulation data is also used to better quantify the role of the latter on the recovery of the wake. The manuscript main conclusion is that the lateral and vertical gradients of Reynolds shear stress are the main mechanism for wake recovery.

I find the manuscript very well written and about a topic of interest for the community. The conclusions, in terms of RANS modelling, are also useful for future developments. I therefore recommend the manuscript for publication with only the following minor remarks:

1. The model relies on a constant eddy viscosity assumption. While it is approximately valid, it is still unphysical as the eddy viscosity cannot be non-zero beyond the turbulent/non-turbulent interface (for the case of a zero-pressure gradient wake with laminar inflow). A correction has therefore been proposed (see A. Townsend, Australian Journal of Chemistry 2, 1949; Cafiero et al., Journal of turbulence, 2020), that predicts radial profiles that follow a modified Gaussian shape. Such profiles differ from a pure Gaussian precisely at the edges of the wake, and can therefore be relevant when assessing the recovery mechanisms of it. While probably this point may not affect the qualitative discussion made in the manuscript, I consider it still should be addressed.

Thank you for these references, we have added them to the article and added the following discussion in text:

[revised manuscript text omitted]

2. While this is a recurrent discussion in the field, and the answer is not clear, the far wake should be better defined. This work covers streamwise distances up to 10 diameters, and therefore the wake recovery mechanisms may change further away, when the velocity deficit scales as a power law.

We are not aware of an exact definition of the far wake. In our opinion, the far wake is the region where the (mean) wake is self-similar. Effectively, this means that the far wake is the region where the wake has forgotten how it was generated and effects of the near wake and/or thrust force distribution are no longer visible. We added our definition of the far wake in the beginning of Section 2:
*"The wind turbine wake can be split into near and far wake regions (Vermeer et al., 2003). The near wake is a result of the wind turbine blade forces and it is characterized by complex vortex structures that break down in to smaller turbulent eddies further downstream. The near wake velocity deficit is mainly a footprint of the wind turbine thrust force distribution and it diffuses downstream in a smoother velocity deficit profile. We define the far wake as the region where the mean velocity deficit has become self-similar. In other words, the far wake has forgotten how it was generated and only information of the total wind turbine extracted momentum is known."*
The available large-eddy simulation data set is not suited to provide statistically converged mean quantities for distances far beyond 7.5 diameters downstream. We do not see a different wake recovery mechanism between 7.5-20 D because the wake has already reached the far wake at 7.5D, although the signal to noise ratio deteriorates due to the wake recovery. In order to investigate the wake much further downstream, i.e. 100 rotor diameters, one could also look at Reynolds-averaged Navier-Stokes simulations of a single wake simulation, where the mean results are directly calculated. Our experience is that those simulations would show a continuing Gaussian lateral streamwise velocity deficit profile at hub height that keeps decreasing in the maximum deficit and continues to expand in the lateral distance as function of the downstream distance (see Appendix A). This also has to be the case because the integral of the velocity deficit must be equal to the wind turbine extracted momentum in order to conserve momentum. In real life conditions, such stationary cases are hard to observe as the atmospheric conditions (wind direction, wind speed and atmospheric stability, etc.) are rarely constant over long periods and distances; again we are considering mean values.

3. While there is some qualitative agreement between figures 1 and 3, the authors never compare them quantitively. First, figure 1 corresponds to which streamwise distance? Furthermore, figures 2 and 3 cannot be compared at 7.5D by matching the maximum velocity deficit?

The simple model is not meant to be used as a predictive model; instead it is only applied to illustrate the main wake recovery mechanisms. Hence, we prefer to not make a quantative comparison between the simple model and the large-eddy simulation results.
The streamwise distance is not a direct parameter in the simple model. As stated in Line 44, the normalized maximum deficit, $\Delta \tilde{U}_{\mathrm{max}}$, could be made a function of the downstream distance. The results in Fig. 1 are made with $\Delta \tilde{U}_{\mathrm{max}} = 0.4$, which could reflect a certain downstream distance. We have added short note when Fig. 1 is introduced:
*"The results in Fig. 1 are made with $\Delta \tilde{U}_{\mathrm{max}} = 0.4$, which could reflect a certain downstream distance, although the overall behavior is not influenced by $\Delta \tilde{U}_{\mathrm{max}}$."*

4. A relevant assumption of the model is the self-similarity of some averaged turbulence quantities. This is an important point for both theoretical and numerical modelling (like shown, for instance, in Johansson et al., Physics of Fluids, 2003). I think it would strengthen the manuscript to comment of the self-similar nature (or not), of the curves from figure 3a.

We have not assumed a self-similarity for the turbulence quantities in the simple model, but their self-similarity is simply a result of the assumptions mentioned in Section 2.
Thank you for the reference to Johansson et al. (2003). They have shown self-similar profiles of the velocity deficit but conclude that higher order moments continue to develop downstream. We added a

comment regarding the self-similarity of the turbulence in Sect. 2:

*"The assumed Gaussian velocity profile of the simple model also results in self-similar shear stresses and stress divergence terms. Johansson et al. (2003) argued that higher order velocity moments of an axi-symmetric wake can be shown to develop downstream over large distances and continue to contain information of the near wake. It remains unclear if this conclusion can be applied to a utility scale wind turbine wake due to the mismatch in Reynolds-number."*